# GRADUATED NON-CONVEXITY FOR ROBUST SELF-TRAINED LANGUAGE UNDERSTANDING

## ABSTRACT

Self-training has been proven to be an efficient strategy for unsupervised fine-tuning of language models using unlabeled data and model-generated pseudo-labels. However, the performance of self-trained models is unstable under different conditions of the training and evaluation data, influenced by both data distribution and pseudo-label accuracy. In this work, we propose an outlier robust self-training method based on graduated non-convexity (GNC) to mitigate the problem. We construct self-training as a non-convex optimization problem with outlier training examples. The models are self-trained with robust cost functions according to Black-Rangarajan Duality. The algorithm learns slack variables as the loss weights for all training samples. The slack variables are used to calibrate the loss items during training to update the model parameters. The calibrated loss items lead to more robust self-trained models against different training and evaluation data and tasks. We conduct experiments on few-shot natural language understanding tasks with labeled and unlabeled data examples. Experimental results show that the proposed loss calibration method improves the performance and stability of self-training on different tasks, benefiting the robustness against incorrect pseudo-labels, imbalanced training data, overfitting, and adversarial evaluation data.

## 1 INTRODUCTION

Recent developments in large-scale pretrained language models has significantly improved the performance of natural language understanding tasks (Devlin et al., 2018; Liu et al., 2019; Clark et al., 2020; He et al., 2020; Brown et al., 2020). After pretraining, these models are typically fine-tuned on task-specific training data with human-generated labels. However, human-generated labels are not available (or large enough) for all tasks of interest. When there is a significant number of unlabeled examples, a pretrained model can utilize techniques such as self-training to improve performance (He et al., 2019; Zoph et al., 2020). An issue with this approach is that the generated pseudo-labels can be noisy (Zhao et al., 2021; Lang et al., 2022; Zhang & Zhou, 2011), since a pretrained model can make wrong predictions for unseen examples.

We propose a learning-based loss calibration strategy that tunes the loss weights for each data example during self-training. In this approach, a subset of training data are assigned low weights for calculating the overall training loss, leading to less influence in parameter updates. To learn the loss weights, we employ the graduated non-convexity (GNC) strategy (Yang et al., 2020) based on Black-Rangarajan Duality (Black & Rangarajan, 1996). Under the fully supervised and outlier-free setting, the model parameters are updated to optimize the total cost of all training samples, while under the self-training setting, the training set usually contains outliers with wrong pseudo-labels. We thus optimize a robust cost function

$$L_r = \sum_i \rho[l(\hat{y}_i, x_i); \theta] \tag{1}$$

where $l(\cdot)$ is the selected loss function and $D_{train} = \{(x_i, y_i) | i \in [0, N]\}$ is the training set, and $\theta$ stands for the set of trainable parameters. $\rho(\cdot)$ is a robust cost function and $\hat{y}_i$ stands for an element of a noisy pseudo label set. Such methods have been widely applied in computer vision (Black & Rangarajan, 1996) and robot perception (Yang et al., 2020) where there exists a ground-truth

optimization target for the algorithm to predict. Few application examples have been seen in other domains and the approach has not been applied in the area of self-training. The difficulty is that unlike prediction, an optimal self-training loss on training examples does not guarantee minimal loss on evaluation sets. In fact, adding noise in self-training can produce better performance, for example with Dropout (He et al., 2019) and confidence regularization (Zou et al., 2019).

In this work, we propose GNC self-training that produces robust performance against incorrect pseudo labels, imbalanced training data, overfitting, and adversarial evaluation data. The core idea is penalizing outliers with wrong pseudo labels and over-confident training samples by assigning low loss weights. As a result, such training samples contribute less to the updated model parameters than other samples. In this work we make the following contributions,

**Applying graduated non-convexity (GNC) in self-training.** To the best of our knowledge, this work is the first attempt of applying GNC for self-trained language models. While traditional GNC methods treat high-loss observations as outliers, we noticed that in self-training, outliers include both high-loss and over-confident examples. We propose a shifted robust cost function to deal with over-confident cases, and the loss weights are tuned during self-training.

**Robustness against different training and evaluation data.** The difference within different training and evaluation corpora leads to significant performance gaps between the two because of the accuracy of pseudo-labels and the difficulty/coverage of data examples. We find that the proposed method is robust against different tasks, training corpora, and adversarial evaluation data, outperforming label-free self-training strategies and label-dependent few-shot learning baselines.

## 2 RELATED WORK

Recent pretrained language models are trained with a self-supervised learning strategy, including predicting masked words (Devlin et al., 2018; Liu et al., 2019; He et al., 2020), predicting next words (Brown et al., 2020; Raffel et al., 2020; Lewis et al., 2019), predicting the correctness of words (Clark et al., 2020), and representing sentences (Gao et al., 2021; Chuang et al., 2022). Besides pretraining on corpora without additional human-generated labels, textual entailment corpora, including SNLI (Bowman et al., 2015) and MultiNLI (Williams et al., 2018) are also used for both sentence-level pretraining (Reimers & Gurevych, 2019) and downstream tasks; for example relation extraction (Obamuyide & Vlachos, 2018; Yin et al., 2019) and fact checking (Thorne & Vlachos, 2018).

While most self-supervised learning methods are agnostic to downstream tasks, self-training models learn to handle any downstream task by learning from synthetic data or pseudo-labels that are automatically generated by the same or other models. Among recent research under the context of self-training, back translation (He et al., 2019) and data augmentation methods (Xie et al., 2020; Chen et al., 2020) use synthetic texts as inputs, pseudo labeling methods (Zoph et al., 2020; Zou et al., 2019) use synthetic labels as training targets, and self-trained question answering methods (Bartolo et al., 2021; Luo et al., 2022) use synthetic questions as inputs and synthetic answers as target outputs.

Recent studies have discussed the efficiency and robustness of language models from different aspects, including different settings on training and evaluation data. The authors of Zang et al. (2019); Jin et al. (2020); Bartolo et al. (2020); Wang et al. (2021) indicated that even state-or-the-art language models make mistakes on adversarial evaluation examples. On the other hand, even for language models pretrained on large corpora, their performance on downstream tasks might not be satisfying if not enough task-specific training data is presented (Schick & Schütze, 2021; Le Scao & Rush, 2021). There is little work simultaneously dealing with the limitations of small training sets and missing human-generated labels.

## 3 BACKGROUND: GRADUATED NON-CONVEXITY

Graduated non-convexity (GNC) is a popular optimization method in vision (Blake & Zisserman, 1987) and machine learning (Rose, 1998; Mobahi & Fisher, 2015). The core idea is optimizing a robust cost function instead of a standard loss function, as shown in Equation 1. The robust cost function $\rho(\cdot)$ is adjustable give a coefficient $\mu$. When $\mu$ is large enough, $\rho(l(y_i, x_i))$ is approximately convex. During the optimization, the value of $\mu$ is decreased at each step and the optimization prob-

lem is gradually recovered to the original non-convex problem. A more convex iteration provides a guess for the following optimization.

A typical robust surrogate cost function is the Geman-McClure (GM) function. The GM function and the controlled GM function are shown as follows, as mentioned in (Yang et al., 2020),

$$\rho(l(\cdot)) = \frac{\bar{c}^2 l^2}{\bar{c}^2 + l^2}; \ \rho_\mu(l(\cdot)) = \frac{\mu \bar{c}^2 l^2}{\mu \bar{c}^2 + l^2} \tag{2}$$

where $\bar{c}$ is a parameter that decides the shape of the GM function, and represents the largest cost of a threshold of outlier losses. According to Equation 2, when $\mu \to \infty$, $\rho_\mu(l) \to l^2$; when $\mu = 1$, $\rho_\mu(l) = \rho(l)$. The optimization result of $\rho(l)$ is reached by optimizing a high-quality guess of $\rho_\mu(l)$.

In practice, following (Yang et al., 2020), the optimization target we apply is a mutation of $\rho_\mu(l)$ based on the Black-Rangarajan Duality (Black & Rangarajan, 1996), which proved that the problem of optimizing $\rho_\mu(l)$ is equivalent to optimizing a weighted sum of $l(\cdot)$ and a weight penalty term $\Phi$. In other words, the optimization problem in Equation 1 has the same optimal solution with the following task,

$$x = argmin_\theta \sum_i [w_i \cdot l(y_i, x_i; \theta)^2 + \Phi(w_i)], w_i \in [0, 1] \tag{3}$$

The set of $w = \{w_1, w_2, \ldots, w_n\}$ are slack variables to control each loss item $l(y_i, x)$. In the robotic perception and computer vision areas, $y_i$ stands for a noisy observation of the $i$-th sensor and $x$ stands for the ground-truth location or shape of the physical object of interest. According to Black & Rangarajan (1996), when the GM function is applied as the robust cost function, the weight penalty function $\Phi(w_i)$ can be written as

$$\Phi_{p_\mu}(w_i) = \mu \bar{c}^2 (\sqrt{w_i} - 1)^2 \tag{4}$$

In this work, we jointly optimize the weighted sum of the standard loss $l(y_i, x)$ and the weight penalty loss $\Phi(w_i)$.

## 4 METHOD: GNC SELF-TRAINING WITH SHIFTED GEMAN MCCLURE

### 4.1 ENTAILMENT PRETRAINING AND SUPPOSITION VERIFICATION

In this work, we investigate weakly supervised domain and task adaption of textual entailment models, where the size of training texts and the access to human-generated labels are limited. To enable the adaptation, we formulate natural language understanding tasks to classifications tasks where a model just needs to predict the truth value of input suppositions. We first pretrain entailment classifiers on MultiNLI (Williams et al., 2018) that learns to predict the truth values of the following suppositions based on a hypothesis {x} and the premise {y} ,

$s_{mnli}^1$: {x} is entailed by {y}; $s_{mnli}^2$: {x} is not true when {y} is true.

According to the definition of entailment, $s_p^1$ and $s_p^2$ are contradictory, thus they cannot both be true of false. A model trained to predict the truth value suppositions constructed by each hypothesis-premise pair $(h_i, p_i)$ can be applied to other tasks, if similar suppositions can be constructed. For example, an entailment classifier can be applied to handle the movie review classification task (SST2) (Socher et al., 2013). A sentiment analysis task can also be formulated as an entailment prediction or supposition verification task, with the following potential suppositions,

$s_{sst2}^1$: I like the movie is entailed by my comment {x}; $s_{sst2}^2$: The movie is good because {x}.

We construct the suppositions following Anonymous (2022) for different tasks, including SST2, RTE (Dagan et al., 2005), QNLI (Rajpurkar et al., 2016), QQP (Wang et al., 2018), and their adversarial evaluation corpora (Wang et al., 2021).

### 4.2 SELF-TRAINING OF THE ENTAILMENT CLASSIFIER

In this work, we investigate learning task-specific models with a pretrained entailment classifier with a limited amount of unlabeled training data $D_{task} = \{x_1, x_2, \ldots, x_K\}$, where $x_i$ stands for an

unlabeled data example. Given a pretrained entailment model $M$, We generated pseudo-labels for each data example by feeding suppositions constructed based on $x_i$,

$$\hat{y}_i = M[s_{task}(x_i)] \tag{5}$$

The pseudo-labels are then used to train task-specific models. A new training set is constructed as $D_{train} = \{(x_1, \hat{y}_1), (x_1, \hat{y}_2), \ldots, (x_K, \hat{y}_K)\}$. The new training set is used to fine-tune the pretrained entailment classifier. Note that the pretrained entailment model has three classes (entailment, neutral, and contradiction), we just use the entailment and contradiction scores for the fine-tuning and evaluation of downstream tasks.

### 4.3 CONVEXITY CONTROLLED TRAINING

The Black-Rangarajan Duality (Black & Rangarajan, 1996) suggests that optimizing the surrogate robust cost function $\sum_i \rho(l(y_i, x))$ is equivalent to optimizing the following cost function

$$L_\rho = \sum_i [w_i \cdot l^2(y_i, x_i; \theta) + \Phi(w_i)], w_i \in [0, 1] \tag{6}$$

In this work, we apply the Geman McClure (GM) function as the robust cost $\rho(\cdot)$. Before the beginning of training, all slack variables $w_i$ are set to 1. We use the the alternating optimization strategy proposed in Yang et al. (2020). At each optimization step $t$, we first update the model parameters $\theta$ based on the weighted sum of base costs,

$$\theta^t = \text{argmin}_\theta \sum_i w_i^{t-1} \cdot l^2(y_i, x_i; \theta) \tag{7}$$

then we fix the learned $\theta$ and update the slack variables $W = [w_1, w_2, \ldots, w_n]$, Note that in Yang et al. (2020) and in related work in visual perception, $n$ stands for the number of sensor observations, but in our work $n$ stands for the number of training examples. At each time step, $w_i$ is updated as

$$w_i^t = \text{argmin}_{w_i} \sum_i [w_i \cdot l^2(y_i, x_i; \theta^t) + \Phi(w_i)] = \left[ \frac{\mu \bar{c}^2}{l^2(y_i, x_i; \theta^t) + \mu \bar{c}^2} \right]^2 \tag{8}$$

This analytic solution is obtained by solving $\frac{\partial L_\rho}{\partial w_i} = 0$ when $\theta^t$ is fixed. The output $w_i^t$ is used to calculate $\theta^{t+1}$ in the next episode. At the first training step, $\mu$ is initialized as $\mu = 2l_{max}^2(y_i, x_i; \theta^0)/\bar{c}^2$. In the following training steps, we decrease it by $\mu = \mu/2.8$. The training stops when $\mu < 1$.

It is straightforward to see that an observation, or training example $(y_i, x_i)$ receives a low weight $w_i$ when its corresponding loss $l(y_i, x_i; \theta)$ is high. In visual perception, this feature helps the algorithm to find outlier sensors. In these tasks, excluding outliers benefits the accuracy of the predictions because 1) the perception functions of the sensors are usually known, and 2) accurate object predictions must lead to low cost if the observation is correct. The situation is different under the context of training neural language models.

In neural language model training, the models are presented with $(y_i, x_i)$ pairs instead of $(y_i, \theta)$, so there is no guarantee if the model tuning step in Equation 7 could find a global optimal solution of $\theta$ under the current setting of $w$ or not. On the other hand, a high-loss example in machine learning is not necessarily an outlier - it might bring new knowledge that improves the model. Another difficulty is that an outlier might also improve the model performance and robustness. Under the self-training setting, a low cost might be associated with either a correct pseudo-label or a wrong pseudo-label that the model is over-confident about. These difficulties prevent the direct application of GNC on machine learning, especially self-training of neural language models.

To mitigate the problem, we introduce the shifted Geman McClure (SGM) function for graduated non-convexity optimization. Besides the parameter $\bar{c}$ that controls the maximum cost of non-outlier examples, we set another parameter $\epsilon$ that bars the minimal normal cost of non-outlier examples. Under this setting, the outlier set consists of two parts, including high-loss examples associated with wrong pseudo-labels, and over-confident examples whose loss is too low that might lead to overfitting of the trained model. The SGM function applied as our robust cost is

$$\rho_\mu(l) = \frac{\mu \bar{c}^2 (l - \epsilon)^2}{\mu \bar{c}^2 + (l - \epsilon)^2} \tag{9}$$

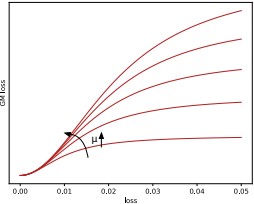 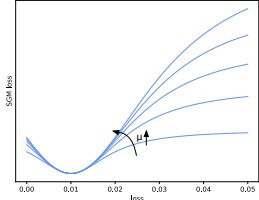

Figure 1: Example of GM (left) and SGM (right) functions under different $\mu$ and loss values.

When $l < \epsilon$, the robust cost $\rho$ increases if $l$ continues to decrease. We use this method to prevent overfitting caused by over-confident examples. The shapes of GM and SGM functions are shown in Figure 1. Higher $\mu$ values result in higher curves.

## 5 EXPERIMENTS

### 5.1 TASK AND DATA

**MultiNLI.** We use the multi-genre natural language inference (MultiNLI) (Williams et al., 2018) corpus for pretraining entailment classifiers. The MultiNLI corpus contains 393k sentence pairs that can be classified into 3 categories: entailed, neutral, and contradictory. We train a classifier to predict the truth value of the constructed suppositions as mentioned in Section 4.1.

**QNLI/AdvQNLI.** The question natural language inference (QNLI) corpus contains question-context pairs. The task is to predict if the answer to a given question is contained in its paired passage. We construct the following supposition for the task, $s_{qnli}$: `The answer to question {q} is entailed by {p}`.

**QQP/AdvQQP.** The Quora question pairs (QQP) task is proposed in Wang et al. (2018). In the QQP task, a model learns to predict if two questions are duplicated. In most cases, the task is to predict if two questions can be answer by the same answer. The adaptation from MultiNLI to QQP is a cross-task adaptation, and the supposition for the QQP task is $s_{qqp}$: `The answer to {q₁} is entailed by the answer to {q₂}`.

**RTE/AdvRTE.** The recognizing textual entailment (RTE) task is a binary (entailment/contradiction) entailment classification task based in the news domain. Each data example is a claim-passage pair and the task is predicting if the claim is entailed by the paired passage or not. The adaptation from MultiNLI to RTE is cross-domain adaptation and the suppositions are the same as MultiNLI suppositions, as defined in Section 4.1.

**SST2/AdvSST2.** The Stanford Sentiment Treebank (SST-2) is a sentimental analysis corpus that requires a model to predict if the attitude of a movie review is positive or negative. The adaptation from MultiNLI to SST-2 is a cross-task adaptation. The construct supposition template is $s_{sst2}$: `The movie is good is entailed by the comment {x}`.

### 5.2 IMPLEMENTATION DETAILS

In this work, we train BERT (Devlin et al., 2018) and DeBERTa He et al. (2020) models as the pretrained entailment classifiers, which are trained on the MultiNLI corpus Williams et al. (2018). In GNC self-training, there are 2 key hyper-parameters: the threshold of outlier losses $\bar{c}$ and the amount of shift in the SGM function $\epsilon$. For each task, we select a value from $\{1e-2, 1e-3, 1e-4\}$ for $\bar{c}$. To find an appropriate $\epsilon$, we calculate the losses $l \in R^n$ on all training samples after each training epoch and set $\epsilon = \mu(l) - \alpha(\sigma(l)/\sqrt{n}), \alpha \in [1.2, 2.8]$. More details are in Appendix B.

### 5.3 EXPERIMENTAL RESULTS

In this work, we investigated different experimental settings, including few-shot self-training and the combination of few-shot fine-tuning and self-training.

| Method | GLUE | | | | Method | AdvGLUE | | | |
|---|---|---|---|---|---|---|---|---|---|
| | QNLI | QQP | RTE | SST2 | | QNLI | QQP | RTE | SST2 |
| Few-shot (left) and supervised (right) medium LMs (350M) with human-generated labels | | | | | | | | | |
| PET | 61.3 | 67.6 | 65.7 | 91.8 | R3F | 47.5 | 40.6 | 50.1 | 38.5 |
| LM-BFF | 69.2 | 69.8 | 83.9 | 90.3 | $CT_T$ | 49.6 | 40.7 | 46.2 | 39.2 |
| P-tuning | 58.8 | 67.6 | 70.8 | 92.6 | MT | 47.5 | 41.5 | 52.5 | 51.3 |
| PPT | 68.8 | 67.2 | 67.9 | 92.3 | BERT | 39.8 | 37.9 | 40.5 | 33.0 |
| UPT | 70.1 | 72.1 | 68.9 | 92.9 | DeBERTa | 57.9 | 60.4 | **79.0** | 57.8 |
| Few-shot large LMs (137B) with human-generated labels | | | | | | | | | |
| LaMDA | 55.7 | 58.9 | 70.8 | 92.3 | \ | - | - | - | - |
| FLAN | 63.3 | 75.9 | 84.5 | **94.6** | \ | - | - | - | - |
| Zero-shot pretrained entailment classifiers based on medium LM (350M) | | | | | | | | | |
| RoBERTa-PT | 71.5 | 78.6 | 81.2 | 87.7 | \ | 62.1 | 52.6 | 61.7 | **59.9** |
| DeBERTa-PT | 77.3 | **79.9** | 84.5 | 90.1 | \ | 61.5 | 64.1 | 66.7 | 42.6 |
| Few-shot Self-trained medium LMs (350M) without human-generated labels | | | | | | | | | |
| RoBERTa-ST | 71.2 | 78.9 | 81.4 | 86.0 | \ | 63.9 | 56.3 | 61.1 | 54.7 |
| RoBERTa-GNC | 77.4 | 79.1 | 80.8 | 88.1 | \ | 66.1 | 60.2 | 65.6 | 48.2 |
| DeBERTa-ST | 76.1 | 79.4 | 82.9 | 89.1 | \ | 63.2 | 64.7 | 69.2 | 47.0 |
| DeBERTa-GNC | 78.7 | 79.8 | 83.8 | 90.4 | \ | 65.3 | **68.6** | 69.0 | 47.5 |
| Few-shot supervised + Self-trained medium LMs (350M) with labeled and unlabeled data | | | | | | | | | |
| RoBERTa-ST | 79.3 | 75.7 | 84.4 | 88.7 | \ | 61.9 | 60.0 | 64.9 | 52.7 |
| RoBERTa-GNC | 80.2 | 76.9 | 84.9 | 90.3 | \ | 62.9 | 60.0 | 64.7 | 57.8 |
| DeBERTa-ST | 79.6 | 78.2 | 85.1 | 90.7 | \ | 65.4 | 64.9 | 74.9 | 47.7 |
| DeBERTa-GNC | **83.1** | 79.4 | **85.5** | 91.1 | \ | **66.5** | 67.5 | 78.1 | 48.7 |

Table 1: Experimental results of few-shot supervised, fully-supervised, and few-shot self-training (unsupervised) models and methods. BERT and DeBERTa results for AdvGLUE are trained with entire training corpora. The self-training results are averaged with 20 independent experiments.

**Few-shot self-training.** Under the few-shot self-training setting, a pretrained entailment classifier can only access a limited number ($K = 12$ or $132$) of unlabeled data examples. The pretrained models generate pseudo labels for fine-tuning. We evaluate pretrained DeBERTa and RoBERTa entailment classifiers on both regular and adversarial evaluation sets. We conduct the evaluation with 20 independent experiments and report the average performance.

**Combined Few-shot fine-tuning and self-training.** In many situations, the model can access a limited number of training examples with human-generated labels and a larger number of unlabeled data examples. In this case, we first fine-tune the pretrained models with the labeled data, and self-train the fine-tuned model using unlabeled data with generated pseudo-labels. We evaluate the regular self-training strategy with GNC self-training with few-shot fine-tuned models. In each experiment, we use 12 human-labeled training examples and 132 unlabeled data examples.

We compare the performance of the proposed methods with strong few-shot and fully supervised baselines. While the baseline models access task-specific human-generated labels, our pure self-training method only process unlabeled data examples from each task. The few-shot learning baselines include PET Schick & Schütze (2021), LM-BFF Gao et al. (2020), P-tuning Liu et al. (2021), PPT Gu et al. (2021), and UPT Wang et al. (2022), which are medium-sized models with around 350M parameters trained with 16 human-labeled examples. FLAN Wei et al. (2021) and LaMDA Thoppilan et al. (2022) are large-scale language models with 137B parameters and are tuned with up to 12 data examples. The robust baselines R3F Aghajanyan et al. (2020), $CT_T$ Xu et al. (2021), and MT Tong et al. (2022) are trained on the full training corpora.

The experiment results are shown in Table 1. We found that the self-training methods receive significant improvement on adversarial evaluation data, while matching or outperforming the pretrained entailment classifiers on regular evaluation sets. We cite the performance of baseline methods reported by the authors, and summarize the self-training performance by averaging the results of 20 independent experiments. We found the self-training methods that are tuned with 12 unlabeled data examples outperform the 16-shot and fully-supervised baselines in most experiments. Comparing

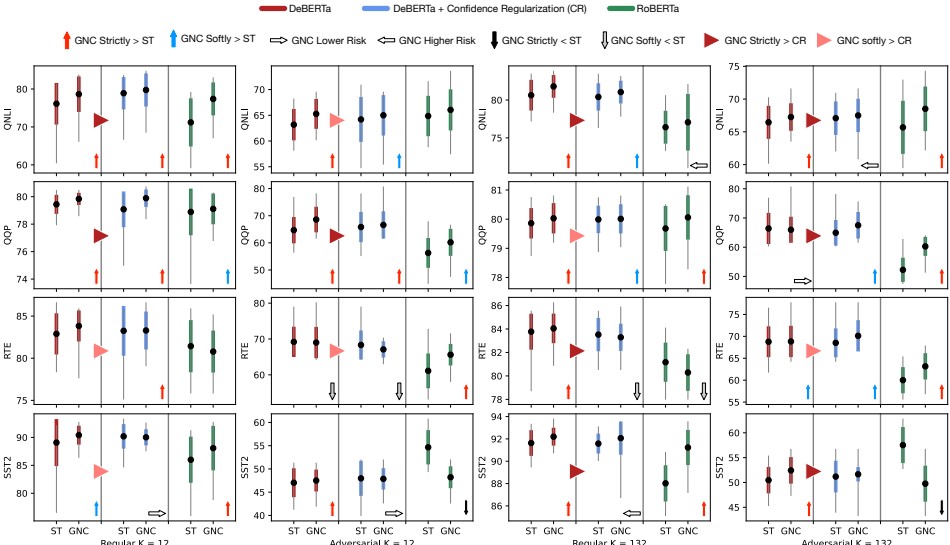

Figure 2: The performance (%) of DeBERTa- and RoBERTa-large models with different self-training strategies under the few-shot self-training setting. The black dots stand for mean performance, the colored bars stand for the range of $\pm 1$ standard deviation, and the grey lines indicate the minimum and maximum accuracy. All results are summarized with 20 independent experiments. ST stands for the regular self-training and GNC stands for self-training with graduated non-convexity. $K$ stands for the number of unlabeled training examples processed by the models.

the best few-shot and fully-supervised results of medium language models, our GNC self-training method improved 18.8%/7.4% on QNLI/AdvQNLI and 7.7%/8.2% on QQP/AdvQQP.

Although GNC constantly outperforms the regular self-training method, its performance is significantly lower than fully-supervised method on AdvRTE and AdvSST2. With 12 labeled and 132 unlabeled data examples, the GNC self-training method achieved the highest RTE and AdvSST2 performance in addition, and the accuracy on AdvRTE and AdvSST2 is boosted to the fully-supervised level (78.1% vs. 79.0%, 57.8% vs. 57.8%). In the self-training experiments, we use all unlabeled training data and pseudo labels. We will analyze the effect of data selection in the next section.

## 6    ANALYSIS AND ABLATION STUDY

**Stability and robustness of self-training.** We characterize the stability of the model performance from different perspectives, including the mean value, standard deviation, and minimum/maximum accuracy with different pretrained models and training strategy. To assess the effect of the proposed method, we propose the following metrics,

- A *strictly more stable* model achieves higher mean and minimum/maximum accuracy, with lower standard deviation.
- A *softly more stable* model achieves higher mean and minimum accuracy.
- A *less risky* model achieves higher minimum accuracy and lower standard deviation.

Under the *few-shot* self-training setting, a pretrained entailment classifier can only access a limited number ($K = 12$) of unlabeled data examples. The pretrained models generate pseudo labels for fine-tuning. We evaluate pretrained DeBERTa and RoBERTa entailment classifiers with the confidence regularization (CR) method proposed in Zou et al. (2019) on both regular and adversarial evaluation sets. We conduct the evaluation with 20 independent experiments and report the mean, minimum, maximum performance, and standard deviation in Figure 2.

The experimental results show the effect of GNC in self-training. In all 48 comparisons in Figure 2, the GNC method is strictly more stable than the baselines in 23 cases, where all metrics (mean, std,

min, max) indicate that GNC perform better. There are 35 (72.9%) cases where GNC is softly more stable or less-risky than its baselines. There are only 2 cases where the baselines are strictly better than GNC (AdvSST2, $K = 12, 132$, RoBERTa) and another 4 cases where the baselines are softly better than GNC (AdvRTE, $K = 12$; RTE, $K = 132$).

| GNC vs ST | DeBERTa | DeBERTA+CR | RoBERTa | Reg. K=12 | Adv. K=12 | Reg. K=132 | Adv. K=132 |
|---|---|---|---|---|---|---|---|
| Strictly Stable | **11** | 4 | 8 | 7 | 5 | 6 | 5 |
| Softly Stable | 2 | **5** | 2 | 2 | 2 | 2 | **3** |
| Less Risky | 1 | **2** | **2** | 1 | 1 | 0 | 1 |
| Total / Num of Exp. | **14/16** | 11/16 | 12/16 | **11/12** | 8/12 | 8/12 | 9/12 |

Table 2: Number of evaluations in each performance category with different models and data.

| GNC vs ST | QNLI | QQP | RTE | SST2 | AdvQNLI | AdvQQP | AdvRTE | AdvSST2 |
|---|---|---|---|---|---|---|---|---|
| Strictly Stable | **4** | **4** | 2 | 3 | 3 | 3 | 2 | 2 |
| Softly Stable | 1 | **2** | 0 | 1 | 1 | **2** | **2** | 0 |
| Less Risky | 0 | 0 | 0 | 1 | 0 | 1 | 0 | 1 |
| Total/Exp. | 5/6 | **6/6** | 2/6 | 5/6 | 4/6 | **6/6** | 4/6 | 3/6 |

Table 3: Number of evaluations in each performance category in different tasks.

We compare the model performance across methods and tasks. We first note that in 15 out of 16 tasks, the performance of DeBERTa+GNC is at least softly more stable than DeBERTa+CR. The only exception is the AdvSST2 task when $K = 12$, where DeBERTa+GNC is less risky than CR as defined. We also found that the DeBERTa-based models generally outperform RoBERTa, except for adversarial QNLI and SST2 evaluations. We also compare the model performance under different model and training data settings. Table 2 indicates that the GNC strategy is more effective on DeBERTa and the improvement is most obvious when $K = 12$. This proves that when the size of training data is limited, self-training is likely to be biased and the performance would be unstable. Note that each experiment stands for the summarized result of 20 independent runs.

The performance analysis across different tasks is presented in Table 3, summarizing all model and training data settings. We found that the QQP and AdvQQP tasks benefits most from the GNC strategy, while the RTE task does not. We believe the reason is that the RTE task is also an entailment classification task, so it is not difficult for the baseline self-training method to adapt.

**Few-shot + self-training performance.** We also analyze the experimental results under the combined few-shot supervised and self-training settings on different tasks, where we first fine-tune the pretrained entailment classifiers with 12 labeled data examples and then conduct self-training with 132 unlabeled examples. The results of RoBERTa and DeBERTa models are visualized in Figure 3. Out of 8 tasks, DeBERTa+GNC is the most stable strategy in general, while RoBERTa+GNC achieves the best performance on the AdvSST2 task. The results indicate that besides the pretrained entailment classifier, the GNC method is also robust for the self-training of few-shot supervised models.

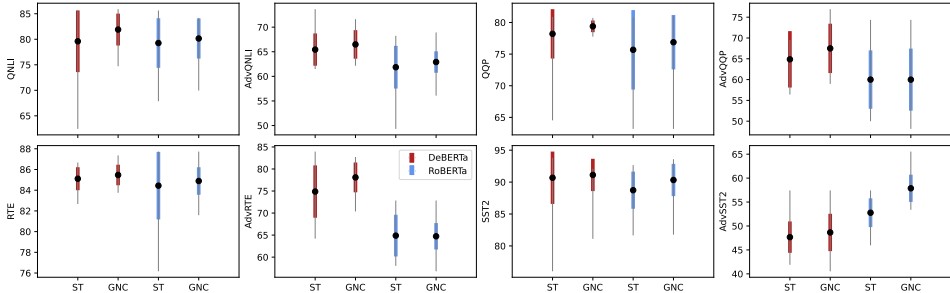

Figure 3: Results of self-training on few-shot supervised models with DeBERTa and RoBERTa using 12 labeled examples and 132 unlabeled examples.

**Geman McClure (GM) and Shifted Geman McClure (SGM) functions.** We analyze the performance of using GM and SGM as the robust cost function on QNLI and SST2 tasks.

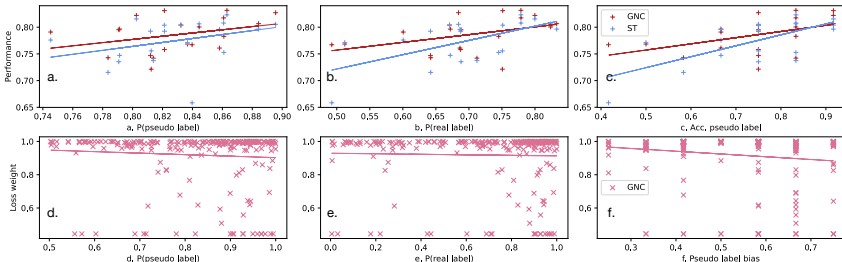

Figure 4: Analysis of the factors that influence self-training performance and the first-epoch weights calculated by the GNC. P(*pseudo label*) stands for the average probability of the predicted pseudo labels, while P(*real label*) stands for the average predicted probability of the human-generated labels. Pseudo label bias stands for the percentage of the pseudo label in all selected data examples.

The experimental results are shown in Table 4. The results indicate that shifting the optimum point of the GM function can lead to more stable performance, by comparing the mean and minimum performance on QNLI and SST2 tasks. However, the variance of GM performance is larger and it achieves higher maximum performance on the adversarial evaluations. Despite this fact, we can still conclude that the SGM cost function introduces more stable and robust self-training performance.

**What Makes Self-training Unstable and How Does GNC Help?** We randomly selected 20 different training sets of the QNLI task, and then conduct the baseline and GNC self-training on each training set. We plot the relation of self-training performance and related factors as shown in Figure 4 with data pairs and linear regressions. From the upper plots, we find that the performance is roughly related to the pseudo-label accuracy and the pseudo-labeling perplexity. Figure 4.a indicates that the confidence of the pseudo-labels affects the self-training performance, and GNC is more robust against the change.

|     |      | QNLI  | AdvQNLI | SST2  | AdvSST2 |
|-----|------|-------|---------|-------|---------|
| GM  | Mean | 75.73 | 64.22   | 89.04 | 46.42   |
|     | Std  | 6.13  | 3.72    | 4.1   | 3.24    |
|     | Min  | 60.28 | 55.41   | 72.71 | 38.51   |
|     | Max  | 82.39 | 70.95   | 92.09 | 52.03   |
| SGM | Mean | 78.67 | 65.27   | 90.4  | 47.5    |
|     | Std  | 4.68  | 2.9     | 1.69  | 2.35    |
|     | Min  | 66.08 | 60.14   | 86.35 | 41.89   |
|     | Max  | 83.8  | 69.59   | 92.78 | 51.35   |

Table 4: Comparison of GM and SGM.

Figure 4.b and c suggest that the accuracy of pseudo-labeling has more impact. When the accuracy labeling accuracy is high ($\sim 90\%$), the baseline and GNC self-training methods achieve similarly high performance. However, when the pseudo-labeling accuracy is low, the performance of the baseline self-training method significantly drops, while the GNC self-training performance is still robust even when the labeling accuracy is lower than $50\%$.

In Figure 4.a, b, and c we plot the loss weights $w_i$ in the first self-training epoch. We find that GNC tends to highlight low-confident data to avoid overfitting as shown in Figure4.d, while it does not highlight all data examples with correct pseudo labels (Figure 4.e). Figure 4.f indicates that when the pseudo label set of the randomly selected data is imbalanced, the data examples of the majority pseudo labels receives lower loss weights. This can mitigate the bias during self-training.

Additional analysis on data selection and adversarial evaluation data is presented in Appendix C.

# 7 CONCLUSION

We apply graduated non-convexity in robust self-training for natural language understanding. With the Black-Rangarajan Duality, we transfer the optimization of a shifted robust cost function to optimizing a weight loss function, where the loss weights are learned to improve the stability and robustness of self-trained language understanding models. Experimental results summarized from 20 independent runs show that the GNC method improves the performance, stability, and robustness of self-training on 4 tasks and corresponding regular and adversarial evaluation corpora. Our label-free method significantly outperforms strong label-dependent baselines. Through the analysis and visualization, we found the GNC self-training strategy is more robust against wrong pseudo-labels, imbalanced training data, overfitting, and adversarial evaluation data.

## ETHICS STATEMENT

In this work we propose a self-training method for language understanding. Traditionally, training a language understanding model needs human annotation on large-scale corpora that contributes to a number of annotation jobs. Our method requires much less human-generated labels, leading to the risk of reducing the number of annotation jobs and pays. On the other hand, since the model learns from its own outputs, it is still possible for the model to generate risky predictions, although the method proposed in this manuscript helps mitigate the risk.

## REPRODUCIBILITY STATEMENT

**Data.** We introduce the tasks and corpora we used for training and evaluation in Section 5.1 and Appendix A. We mentioned how we use the data in Section 5.3.

**Method.** We introduce the background of the method in Section 3, the details of our method in Section 4, and present the pseudo-code of our algorithm in Appendix B.2.

**Hyper-parameter.** We describe the key hyper-parameters of GNC self-training in Section 5.2, and list other regular hyper-parameters in Appendix B.1.

**Experiments.** We describe the experiment results, number of independent runs, and visualized analysis in Section 5.3, 6, and Appendix C to prove the statistical significance.

**Source code.** We will open-source the source code for reproducing the experiment results. The code is currently hosted anonymously on `https://anonymous.4open.science/r/ GNC-ST-DC8F/readme.md`.

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

## A    DATA AND TASK

**MultiNLI.** We use the multi-genre natural language inference (MultiNLI) (Williams et al., 2018) corpus for pretraining entailment classifiers. The MultiNLI corpus contains 393k sentence pairs that can be classified into 3 categories: entailed, neutral, and contradictory. The entailment relation defined in the MultiNLI task is not strict logical entailment. The textual entailment is defined by "the hypothesis is *likely* to be true when the premise is true." The contradiction means "the hypothesis cannot be true when the premise is ture", and when the truth value of the hypothesis cannot be inferred by the premise, the text pair is classified as neutral. We train a classifier to predict the truth value of the constructed suppositions as mentioned in Section 4.1.

**QNLI/AdvQNLI.** The question natural language inference (QNLI) corpus contains question-context pairs. The task is to predict if the answer to a given question is contained in its paired

passage. The regular validation set of QNLI contains 5.6k question-passage pairs for evaluation. AdvQNLI is the adversarial evaluation set proposed by Wang et al. (2021), containing examples generated by textual adversarial attack. The AdvQNLI evaluation set contains 0.9k text examples. The adaptation from MultiNLI to QNLI is a cross-task adaptation. The supposition for the task is

$$s_{qnli}: \text{The answer to question } \{\texttt{q}\} \text{ is entailed by } \{\texttt{p}\}.$$

**QQP/AdvQQP.** The Quora question pairs (QQP) task is proposed in Wang et al. (2018). In the QQP task, a model learns to predict if two questions are duplicated. In most cases, the task is to predict if two questions can be answer by the same answer. The regular evaluation set of QQP contains 391k question-question pairs and the adversarial set contains 0.4k examples. The adaptation from MultiNLI to QQP is a cross-task adaptation, an the supposition for the QQP task is

$$s_{qqp}: \text{The answer to } \{q_1\} \text{ is entailed by the answer to } \{q_2\}.$$

**RTE/AdvRTE.** The recognizing textual entailment (RTE) task is a binary (entailment/contradiction) entailment classification task based in the news domain. Each data example is a claim-passage pair and the task is predicting if the claim is entailed by the paired passage or not. The different between RTE and MultiNLI is that the texts of RTE is generally longer than MultiNLI. The regular evaluation set of RTE contains 3k text pairs, and the adversarial evaluation set contains 0.3k data examples. The adaptation from MultiNLI to RTE is cross-domain adaptation and the suppositions are the same as MultiNLI suppositions, as defined in Section 4.1.

**SST2/AdvSST2.** The Stanford Sentiment Treebank (SST-2) is a sentimental analysis corpus that requires a model to predict if the attitude of a movie review is positive or negative. The regular evaluation set contains 1.8k test reviews and the adversarial evaluation set contains 1.4k examples. The adaptation from MultiNLI to SST-2 is a cross-task adaptation with a gap between input formats. While the inputs of all tasks described above are text pairs, the inputs of SST-2 are movie reviews without any pair text. To apply the pretrained entailment classifier, we construct both label descriptions and suppositions. The construct supposition template is

$$s_{sst2}: \text{The movie is good is entailed by the comment } \{x\}.$$

# B ADDITION IMPLEMENTATION DETAILS

## B.1 HYPERPARAMETER TUNING

The models are pretrained and fine-tuned with the AdamW optimizer Loshchilov & Hutter (2017). In pretrained, the learning rate is set to $5e$-6 for BERT and $3e$-6 for DeBERTa, and the weight decay weight is set to $1e$-5. For the downstream self-training step, the learning rate for both models is set to $4e$-6 and the weight decay is $1e$-2. The learning rates and weight decay coefficients are fixed across all tasks. We tune the baseline self-training models and found that the best epoch number is 6. In GNC self-training, the model automatically decides the epoch number.

## B.2 GNC IMPLEMENTATION

We summarize the process of GNC self-training with pseudo code in Algorithm 1.

# C ADDITIONAL ANALYSIS

**Data selection.** Lang et al. (2022) have mentioned that selecting high-quality data-pseudo label pairs can benefit self-training. We hereby analyze the effect of model confidence-based data selection (CS) on all tasks we investigate in this work. The experiment results are presented in Figure 5.

We noticed that under our setting where a limited number of unlabeled data is used for self-training, the data selection strategy does not guarantee improvement on all tasks. However, we found the the GNC method can also benefit the data selection strategy on all tasks except AdvRTE, improving the average performance of 20 experiments and reduce the risk of getting low performance. On QQP, RTE, and SST2, GNC self-training without data selection is still the most robust strategy.

---

**Algorithm 1** GNC self-training with shifted Geman McClure (SGM)

---

**Require:** Pseudo training set $D = \{(x_0, y_0), (x_1, y_1), \ldots, (x_n, y_n)\}$, model $M_0, \bar{c}, \epsilon$

    $\delta \leftarrow 2.8$                                                    ▷ update step size of $\mu$

    $\mu \leftarrow 100$

    $t \leftarrow 0$                                                        ▷ number of steps

    $[w_0, w_1, \ldots, w_n] \leftarrow [1, 1, \ldots, 1]$

    **while** $\mu > 1$ **do**

        $l_t^i = \texttt{loss\_fn}(M_t(x_i), y_i)$                            ▷ Calculate training loss

        $L_t = \sum_i l_t^i \cdot w_i$                       ▷ Calculate the weighted total loss

        $M_{t+1} \leftarrow \texttt{train}(M_t, L_t)$        ▷ update model parameters using the total loss

        $e_t^i = \texttt{loss\_fn}(M_{t+1}(x_i), y_i)$        ▷ calculate evaluation loss of training data

        **if** $t == 0$ **then**

            $\mu = 2 \cdot \texttt{max}_i e_i^2 / \bar{c}^2$                               ▷ initialize $\mu$

        **else if** $t > 0$ **then**

            $\mu \leftarrow \mu / \delta$                                             ▷ update $\mu$

        **end if**

        $w_i = \left[ \frac{\mu \bar{c}^2}{(e_i - \epsilon)^2 + \mu \bar{c}^2} \right]^2$                                ▷ update $w_i$

    **end while**

---

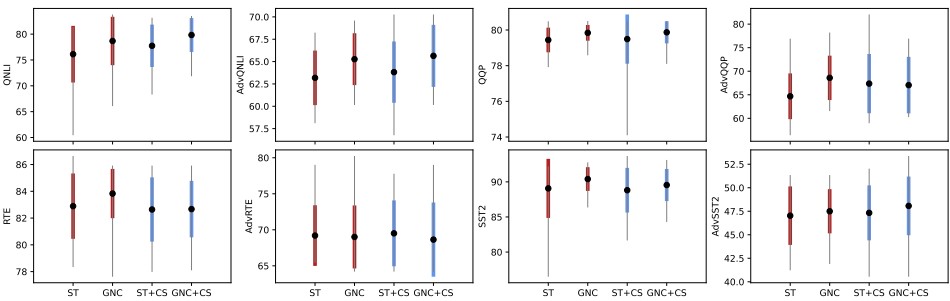

Figure 5: Ablation study of applying confidence-based data selection (CS) on baseline and GNC self-training methods on each task with the DeBERTa model.

**Adversarial evaluation analysis.** We visualize the experiment results with baseline and GNC self-training on AdvQNLI. The results are shown in 6. As the regular evaluation set, the GNC method is also more robust against high pseudo label noises and labeling perplexities.

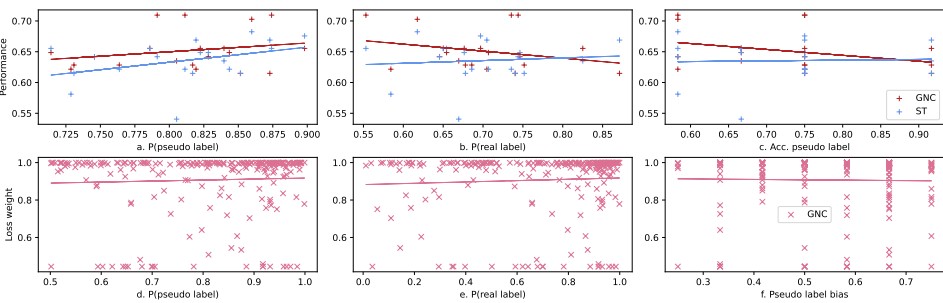

Figure 6: Analysis of 20 experiments on AdvQNLI with the baseline and GNC self-training.

