# OpenReview forum: "Graduated Non-Convexity for Robust Self-Trained Language Understanding"
_ICLR.cc/2023/Conference — Submitted to ICLR 2023_

### Official Review · Reviewer_ZPmQ · 2022-10-23

**Confidence:** 4
**Clarity, Quality, Novelty And Reproducibility:** See above
**Correctness:** 3
**Technical Novelty And Significance:** 1
**Empirical Novelty And Significance:** 2
**Recommendation:** 3

**Strength And Weaknesses:**

Strength:

* This paper considers self-training, which is a very important topic and has potentially wide usage.

* The proposed method is easy to understand, although there are still some presentation issues.

* Experimental results seem to demonstrate the effectiveness of the proposed method.


Weaknesses:

* Some parts of the paper are not clear.

  * In Section 4.1, the authors mentioned they pre-train a model on MNLI. How is the pre-training conducted? Do you train a model with 3-way classification loss (entailment, neutral, contradiction) using the provided labels in MNLI? Or do you train a model to predict $s^1_{\text{mnli}}$?

  * Continuing on the previous comment, how is $s^2_{\text{mnli}}$ related to $s^1_{\text{mnli}}$? You mentioned these two are contradictory, could you give a concrete example? Also, how do you use $s^2_{\text{mnli}}$ in the pre-training stage?

  * What is $\ell_{\text{max}}$ in the paragraph below Eq. 8?


* Some heuristics seem rather arbitrary.

  * For example, in the paragraph below Eq. 8, the schedule of $\mu$ is defined as $\mu \leftarrow \mu/2.8$. Could the authors provide some intuition about this $2.8$ factor? Or is this purely based on trial and error? And why is training stopped when $\mu<1$?

  * Also, in Section 5.2, could the authors provide some intuition about the schedule of $\epsilon$? In particular the additional hyper-parameter $\alpha \in [1.2, 2.8]$?

* The technical novelty and contribution is quite limited. The authors simply introduce an additional hyper-parameter $\epsilon$ into the GNC method. The design of this hyper-parameter is based on intuition, and absolutely no theoretical guarantee is provided,  (which I think is fine if strong enough empirical evidence is provided).

* The authors mention that “we train BERT and DeBERTa models as the pretrained entailment classifiers”. During the entailment pre-training stage, is the model initialized from pre-trained BERT/DeBERTa? If so, the model already contains information that can help downstream tasks such as SST-2.

* The authors mentioned that “we investigate weakly supervised domain and task adaption of textual entailment models”. The authors could run experiments in a more general setting. For example, directly fine-tune on GLUE where only a small proportion (e.g., 10%) of the labels are considered available.

I will raise my score if the authors can address my concerns and run more experiments.

**Summary Of The Paper:**

In this paper, the authors propose an outlier robust self-training method based on graduated non-convexity (GNC) to mitigate the instability issue in conventional self-training. Specifically, they propose new robust cost functions according to Black-Rangarajan Duality. Experimental results on natural language understanding demonstrate the effectiveness of the proposed method.

**Summary Of The Review:**

See above

---

> ### Author Response · Authors · 2022-11-18
> **Will add more details if accepted, but some information already in the manuscript.**
>
> Thank you for your insightful review! We appreciate that you mentioned the importance of the task and the improvement on language understanding results.
>
> About the weakness you mentioned,
> 1. [About MNLI classifier - they are 3-way classifiers]
>
> Yes, we pretrained the model to perform a 3-way classification on MNLI with the [CLS] embedding. We train the model with both $s_{mnli}^1$ and $s_{mnli}^2$. We did not list much detail, but we cited (Anonymous 2022) in section 4.1. due to space limitations. We will add more details on this if accepted.
>
> 2. [About pertaining suppositions - more details are in Anonymous 2022 and will be added if accepted]
>
> Contradictory supposition means that when $s_1$ is true, $s_2$ must be false. As a result, we use both $s_1$ and $s_2$ for training on the same data example, but they are assigned opposite labels (**True** -> **False**). When the original label is neutral, both $s_1$ and $s_2$ are **neutral**. For example, a data example is
>
> - premise - I went to school this morning.
> - hypothesis - I stayed at home this morning.
> - Label - contradiction.
>
> and the constructed suppositions are
> - $s_1$ - I stayed at home this morning is entailed by I went to school this morning (**False**)
> - $s_2$ - I stayed at home this morning is not true when I went to school this morning is true (**True**)
>
> We put both s1 and s2 into the training set and randomly shuffle them to train a 3-way classifier. For the downstream binary classification tasks, we discard the neutral score. Again, more details are in (Anonymous 2022) as we cited.
>
> 3. [About $L_{max}$]
>
> $L_{max}$ stands for the largest loss of all data examples in the training set after a training epoch.
>
> 4. [About hyper-parameter $\mu$ - prevent too long training]
>
> The hyper-parameter is proposed by Yang et al 2020 and the value selected in that paper is 1.4. We selected a larger value to prevent the training to take too many epochs.
>
> 5. [About Black-Rangarajan Duality and SGM with $\epsilon$ - does not affect the Black-Rangarajan Duality]
>
> Introducing the hyper-parameter does not affect the Black-Rangarajan Duality because we can consider optimizing the whole $L-\epsilon$ loss item. Since $\epsilon$ is a constant value, it is equivalent to optimizing $L$.
>
> 6. [About pretrained BERT/DeBERTa models - standard approach, same as famous and cited baselines]
>
> We use the pretrained BERT/DeBERTa models as well as all baseline models, which is a quite standard approach applied in many famous weakly supervised NLP papers, especially the baseline models we compared with.
>
> \*BERT\* based text classification models do not contain information about the downstream tasks because they are only trained for masked language modeling and they do not see task-specific training/evaluation corpora during pretraining. Additionally, they have randomly initialized classification heads needed to be trained before fine-tuning on downstream tasks.
>
> 7. [More general few-shot learning results - have been shown in Table 1 and more will be added if accepted]
>
> We did report experiments with the more general few-shot learning setting. Many famous NLP papers use only K = 12/16/24 labels for few-shot learning, including all baseline models we compared with. In the 5th section in Table 1 (Few-shot supervised + self-trained), we list the performance of using 12 human labels + 132 unlabeled data samples.
>
> We thank the reviewer again for the insightful comments! The reviewer asked about important details in the paper, and we will highlight them with more space if the paper is accepted.

---

### Official Review · Reviewer_HQq2 · 2022-10-24

**Confidence:** 2
**Correctness:** 3
**Technical Novelty And Significance:** 3
**Empirical Novelty And Significance:** 3
**Recommendation:** 6

**Clarity, Quality, Novelty And Reproducibility:**

The clarity is good. The Quality is fair. The Novelty is good. The Reproducibility is good.

**Strength And Weaknesses:**

Strengths:

(1) Novelty of firsly introducing the GNC into self-training to achieve robustness;

(2) Easy understanding for the suppositions construtions which is aimed at formulate different tasks as uniform classification task.

(3) Great performance which validates the effectiveness of the method.

Weakness:

(1) why the coefficient μ is decreased by 2.8 in Eq. 8? As the preliminary introduced that "A more convex iteration provides a guess for the following optimization." Thus the setting of the decreasing of coefficient should be discussed.

(2) The robustness against incorrect pseudo-labels, imbalanced training data, overfitting, and adversarial evaluation data have not been clearly verified in the experiments.


**Summary Of The Paper:**

Facing the problem that self-trained models usually exhibit unstable performance influenced by data distribution and pseudo-label accuracy, this paper firstly applies the graduated non-convexity in self-trainng.
The authors firstly formulate the NLU tasks to classification tasks and construct supposition for different tasks. Then the suppositions are fed into the pre-trained model to generate pseudo-labels for each data example. As the core, authors proposed the convexity controlled training. The method achieves great improvement on downstream tasks.


**Summary Of The Review:**

Please see the comments in Strength And Weaknesses, but I am not an expert in NLP domain, some of my adjustment maybe not correct

---

> ### Author Response · Authors · 2022-11-18
> **The claims in the abstract are supported by AdvGLUE in Table 1 and analysis in Figure 4.**
>
> Thank you for your insightful review! We appreciate that you mentioned the novelty and performance improvement in this work.
>
> About the weakness you mentioned,
> 1. [About hyperparameter $\mu$]
>
> $\mu$ is a hyper-parameter that can be tuned, which is originally proposed by Yang et al [1]. In [1], the selected $\mu$ is 1.4, but in this work, we tuned it and found that 2.8 works fine and prevents the training to take too many epochs. The algorithm optimizes a more convex loss function to find a rough solution, and then decreases to the original non-convex loss and further optimizes. Thus /mu decreases during training.
>
> 2. [About our claim about The robustness against incorrect pseudo-labels, imbalanced training data, overfitting, and adversarial evaluation data]
>
> We evaluated the robustness against adversarial test data with the AdvGLUE benchmark in Table 1 and Figure 2. We also demonstrate the effect of the proposed model against incorrect pseudo labels, imbalanced training data, and overfitting in Figure 4. Figure 4 shows that our method performs better when the pseudo-label accuracy is low and the data is imbalanced. This also indicates that with the proposed method does not overfit to the wrong pseudo labels that hurt the model performance.
>
> [1] Heng Yang, Pasquale Antonante, Vasileios Tzoumas, and Luca Carlone. Graduated non-convexity for robust spatial perception: From non-minimal solvers to global outlier rejection. IEEE Robotics and Automation Letters, 5(2):1127–1134, 2020.

---

### Official Review · Reviewer_py5v · 2022-10-25

**Confidence:** 3
**Correctness:** 3
**Technical Novelty And Significance:** 3
**Empirical Novelty And Significance:** 2
**Recommendation:** 6

**Clarity, Quality, Novelty And Reproducibility:**

The paper is well-written and the use of graduated non-convexity in self-training looks novel. The source code is available.

**Strength And Weaknesses:**

Strength

1. The paper is well-written and the method is clearly presented.

2. Though the main technique is mostly adapted from previous work [1], the use of graduated non-convexity in the self-training setting looks novel to me.

3. The experiments are extensive and the ablation study looks solid.

Weaknesses

1. The empirical improvements compared to baselines are not very stable and to some extent marginal in some datasets.

[1] Heng Yang, Pasquale Antonante, Vasileios Tzoumas, and Luca Carlone. Graduated non-convexity for robust spatial perception: From non-minimal solvers to global outlier rejection. IEEE Robotics and Automation Letters, 5(2):1127–1134, 2020.


**Summary Of The Paper:**

This paper proposes a self-training method based on graduated non-convexity in order to be robust to outliers with wrong pseudo labels and over-confident training samples. Intuitively the core idea of the method is to penalize these two kinds of samples by assigning low loss weights to them. Experimental results show that the proposed method improves the performance and stability of self-training on many different tasks.

**Summary Of The Review:**

Given the strength and weaknesses, I tend to rate the paper as marginally above the acceptance.

---

> ### Author Response · Authors · 2022-11-18
> **The proposed method improves the performance on most tasks and settings**
>
> Thanks for your insightful review! We appreciate that you noticed the novelty and improvement of this work.
>
> About the weakness you mentioned, we evaluated the portion and significance of different tasks in Tables 2 and 3. We found that although the method does not get a significant improvement on all tasks and models, it works well on most of them. For example, the DeBERTa model achieves significant improvement in 87.5% of experiment settings and tasks.
>
> Thank you again for your comments!

---

### Official Review · Reviewer_uzE6 · 2022-10-29

**Confidence:** 4
**Correctness:** 3
**Technical Novelty And Significance:** 2
**Empirical Novelty And Significance:** 2
**Recommendation:** 3

**Clarity, Quality, Novelty And Reproducibility:**

Apart from some confusing points, the paper is overall clear. I believe the method can be reproduced from the code.

**Strength And Weaknesses:**

The paper is overall well-written, with clearly stated contributions. Applying the GNC in self-training is an interesting idea.

Weaknesses:
1.There seems to be many ad-hoc factors introduced into the proposed algorithm. First, changing the GM to SGM cannot ensure that the Black-Rangarajan Duality still holds. In this sense, it is hard to say the SGM loss and the reweighted classification loss are still related. Where does decreasing μ by μ = μ/2.8 come from? Is it 2.8 is the number with the best performance found with experiments?
2.Lacking more comparison with baselines. It is well known that standard self-training is not the perfect solution. There are a lot of works such as FixMatch, which can really improve standard self-training by a large margin. The authors should also include these methods in their comparison.
3.The experiment setup section is confusing. Why some tasks have 16 labeled examples while some have 12? The backbones of different methods are also different, make it very hard to compare except the last two rows.
4.The authors mentioned running each experiment for 20 times and report the mean. Why not report the standard deviation? Is it the stddev is larger than the margin (I’m inferring this from table 4).


**Summary Of The Paper:**

This paper introduces a graduated-non-convexity-based method to improve the fine-tuning performance on NLP tasks. Based on a gradually convexified loss, the models are assigned different weights to the examples in self-training to mitigate the effect of incorrect labels. Experiments indicate the proposed method outperforms standard self-training.

**Summary Of The Review:**

Although the idea in this paper is interesting, the weaknesses make it unsatisfactory for iclr as a top venue. Thus, I tend to reject this work.

---

> ### Author Response · Authors · 2022-11-18
> **Textual self-training is different from the vision domain. Our results and comparisons are solid.**
>
> Thanks for your comment! We appreciate that you noticed the performance improvement led by our method. To respond to the weakness you mentioned,
>
> 1. [About the Black-Rangarajan Duality and $\mu$]
>
> The Black-Rangarajan Duality still holds if we consider optimizing the loss $L-\epsilon$. Since $\epsilon$ is a constant value, it does not affect the optimization of the original loss $L$. The $\mu$ value 2.8 is a hyperparameter. Yang et al. found that the best /mu value of GNC is 1.4, but we slightly tuned the hyper-parameter to prevent the self-training to take too many epochs.
>
> 2. [About other self-training methods]
>
> There are a number of self-training methods in the vision domain, including FixMatch. However, visual self-training methods are very difficult for training language understanding models, since augmentation for text is much more difficult - changing a word might completely change the meaning of a sentence. However, we still compared our method with confidence regularization for self-training in Figure 2 as a strong baseline.
>
> 3. [About the number of training samples and backbone]
>
> Selecting K=12/16/24 shots are all very common in few-shot NLP research. We cite the baseline performance from corresponding papers, which uses 16 training cases. We use 12 training cases following previous work on this topic. However, we significantly outperform the 16-shot baselines with our 12-shot model. As we mentioned in section 5.3, all baseline models have the same (350M) or larger (137B) pretrained language models (BERT, RoBERTa, LaMDA, FLAN) so the comparison is fair.
>
> 4. We did report the stddev in Figures 2, 3, and 5.
>
> Thanks again for the review! We would like to add - a contribution that the reviewer did not mention is that our self-trained 350M-parameter models outperform supervised large-scale 137B-parameter models, which is a new breakthrough.

---

### Decision · Program_Chairs · 2023-01-20

**Decision:**

Reject

**Justification For Why Not Higher Score:**

by reviewer scores and comments. The reviewers provided valid points that were not fully addressed and no reviewers felt strongly about accepting.

**Justification For Why Not Lower Score:**

N/A

**Metareview: Summary, Strengths And Weaknesses:**

Reviewers found the paper well-written and the introduction of Graduated Non-Convexity to self-learning novel and interesting.
They (uzE6, py5v) found the empirical results to be weak or inconsistent over baselines. The authors did report standard deviations, but they show that the improvement in the mean is rarely greater than 1 standard deviation. The authors did not address this main objection.

Several reviewers also objected to the use of heuristic adjustments for $\mu$ and the authors provided a citation but also did not seem to fully address this.